# A common *cis*-regulatory variant impacts normal-range and disease-associated human facial shape through regulation of *PKDCC* during chondrogenesis

Jaaved Mohammed[1], Neha Arora[1], Harold S Matthews[2,3], Karissa Hansen[4], Maram Bader[1], Susan Walsh[5], John R Shaffer[6,7], Seth M Weinberg[6,7,8], Tomek Swigut[1], Peter Claes[2,3,9,10], Licia Selleri[4], Joanna Wysocka[1,11,12]*

[1]Department of Chemical and Systems Biology, Stanford University School of Medicine, Stanford, United States; [2]Department of Human Genetics, KU Leuven, Leuven, Belgium; [3]Medical Imaging Research Center, University Hospitals Leuven, Leuven, Belgium; [4]Program in Craniofacial Biology, Department of Orofacial Sciences and Department of Anatomy, Institute of Human Genetics, Eli and Edythe Broad Center of Regeneration Medicine and Stem Cell Research, University of California, San Francisco, San Francisco, United States; [5]Department of Biology, Indiana University Indianapolis, Indianapolis, United States; [6]Center for Craniofacial and Dental Genetics, Department of Oral and Craniofacial Sciences, University of Pittsburgh, Pittsburgh, United States; [7]Department of Human Genetics, University of Pittsburgh, Pittsburgh, United States; [8]Department of Anthropology, University of Pittsburgh, Pittsburgh, United States; [9]Department of Electrical Engineering, ESAT/PSI, KU Leuven, Leuven, Belgium; [10]Murdoch Children's Research Institute, Melbourne, Australia; [11]Department of Developmental Biology, Stanford University School of Medicine, Stanford, United States; [12]Howard Hughes Medical Institute, Stanford University School of Medicine, Stanford, United States

*For correspondence:
wysocka@stanford.edu

**Abstract** Genome-wide association studies (GWAS) identified thousands of genetic variants linked to phenotypic traits and disease risk. However, mechanistic understanding of how GWAS variants influence complex morphological traits and can, in certain cases, simultaneously confer normal-range phenotypic variation and disease predisposition, is still largely lacking. Here, we focus on *rs6740960*, a single nucleotide polymorphism (SNP) at the 2p21 locus, which in GWAS studies has been associated both with normal-range variation in jaw shape and with an increased risk of non-syndromic orofacial clefting. Using in vitro derived embryonic cell types relevant for human facial morphogenesis, we show that this SNP resides in an enhancer that regulates chondrocytic expression of *PKDCC* - a gene encoding a tyrosine kinase involved in chondrogenesis and skeletal development. In agreement, we demonstrate that the *rs6740960* SNP is sufficient to confer chondrocyte-specific differences in *PKDCC* expression. By deploying dense landmark morphometric analysis of skull elements in mice, we show that changes in *Pkdcc* dosage are associated with quantitative changes in the maxilla, mandible, and palatine bone shape that are concordant with the facial phenotypes and disease predisposition seen in humans. We further demonstrate that the frequency of the *rs6740960* variant strongly deviated among different human populations, and that the activity of its cognate enhancer diverged in hominids. Our study provides a mechanistic explanation of how a common SNP can mediate normal-range and disease-associated morphological variation, with implications for the evolution of human facial features.

## Editor's evaluation

This provides an important exemplar of a coordinated body of work (using differentiated human induced pluripotent stem cells [iPSCs] and phenotyping of a mouse mutant) to dissect the mechanism by which a candidate human single nucleotide variant (SNV) may influence both shape variation in the oro-mandibular region, and susceptibility to cleft lip/palate. The extensive use of genome editing to introduce both targeted deletions and single nucleotide variants, taking into account that each allele has a different natural haplotype background associated with a differing functional readout, is especially compelling. As well as being of general interest to craniofacial biologists, the iPSC targeting approach is more broadly applicable, provided that a relevant functional readout can be identified.

## Introduction

The face is one of the most complex structures of the human body and has changed dramatically over the course of human evolution (*Mitteroecker et al., 2004*). In addition to facilitating the coordination of multiple sensory organs, protection of the brain, and a number of biological adaptations, the human craniofacial complex is a target of sexual selection and plays key roles in communication and other social interactions (*Guo et al., 2014*; *Sheehan and Nachman, 2014*). The human face shape varies tremendously across individuals, much of which is genetically encoded, as manifested by the high similarity of facial features in monozygotic twins and by familial resemblances [reviewed in *Naqvi et al., 2022*]. Recently, much progress has been made in understanding the genetic underpinnings of the normal-range facial variation in human populations through GWAS [reviewed in *Weinberg et al., 2019*; *Naqvi et al., 2022*]. Collectively, these studies have uncovered over 300 loci associated with different aspects of facial shape, indicating that - perhaps not surprisingly - facial shape is a highly polygenic trait [reviewed in *Naqvi et al., 2022*.] Furthermore, as has been seen for many other human phenotypic traits, facial shape GWAS variants map to the non-coding parts of the genome and are especially enriched within *cis*-regulatory elements, such as enhancers (*Claes et al., 2018*). Enhancers are modular genetic elements that activate the expression of their target genes over long genomic distances and in a tissue-specific manner [reviewed in *Shlyueva et al., 2014*; *Heinz et al., 2015*; *Long et al., 2016*; *Furlong and Levine, 2018*]. As would be expected, enrichment of facial GWAS signals in enhancers is specific to cell types relevant for facial morphogenesis during development, including early fetal facial tissue, in vitro derived embryonic facial progenitors called cranial neural crest cells (CNCCs), and their more differentiated descendants, such as cranial chondrocytes (*Claes et al., 2018*; *Naqvi et al., 2021*; *White et al., 2021*). These observations support the *cis*-regulatory origins of normal-range human facial diversity, whereby non-coding genetic variants affect enhancer function and lead to quantitative changes in gene expression in cell types contributing to facial morphogenesis during development.

In addition to the genetic underpinnings of normal-range facial variation, GWAS of non-syndromic cleft lip/palate (nsCL/P) has revealed the polygenic architecture of this common birth defect, with risk variants again mapping mostly to the non-coding parts of the genome (*Leslie, 2022*; *Naqvi et al., 2022*; *Weinberg, 2022*). While CL/P co-occurs with several Mendelian craniofacial syndromes, the majority of cases (~70%) are non-syndromic (meaning, occurring in the absence of other manifestations) and characterized by complex inheritance. We and others previously noted an overlap between sequence variants associated with nsCL/P risk and normal-range face shape variation (*Boehringer et al., 2011*; *Indencleef et al., 2018*; *Indencleef et al., 2021*). For example, the *rs6740960* variant at the 2p21 locus affects the normal-range jaw shape and confers nsCL/P risk in Europeans (*Ludwig et al., 2017*; *Claes et al., 2018*). Interestingly, such SNPs at the intersection of normal and dysmorphic facial features show high heterozygosity in the analyzed populations and affect diverse aspects of the normal-range facial morphology, such as the shape of the jaw, philtrum, or nose (*Indencleef et al., 2018*). These observations raise a question of how such common genetic variants simultaneously confer disease predisposition and normal-range phenotypic variation of distinct facial regions.

Despite the enormous progress in understanding the genetics of the normal-range and disease-associated phenotypic variation of facial features and other human traits, functional follow-up studies have lagged behind. Such studies may involve the discovery of functional variants directly relevant for gene expression, identifying the specific cell types and spatiotemporal contexts they act in, and

pinpointing the genes they regulate [for example: *Sekar et al., 2016*; *Small et al., 2018*; *Caliskan et al., 2021*; *Sinnott-Armstrong et al., 2021*]. Even more rudimentary is our understanding of mechanisms by which changes in gene dosage associated with common genetic variation translate into complex human phenotypes. This scarcity of mechanistic discernment is not unique to the face, but emerges as a general bottleneck in translating GWAS results to genuine insights into molecular and cellular processes underlying complex traits and diseases, and utilizing them for new therapeutic strategies (*Cano-Gamez and Trynka, 2020*).

Here, we deploy an integrated 'SNP-to-phenotype' approach, combining trait-relevant in vitro human stem cell models with in vivo studies in mice to probe the mechanism by which a specific GWAS-identified variant, *rs6740960*, affects both normal-range facial shape and confers nsCL/P risk. We identify a craniofacial enhancer harboring the *rs6740960* SNP, characterize its cell-type specific and facial-domain restricted activity pattern, and describe changes in activity during hominid evolution. Using in situ chromatin conformation capture experiments, we link the *rs6740960* cognate enhancer to its target gene, *PKDCC*. Through genome editing, we then uncover the chondrocyte-specific impact of this enhancer and of the *rs6740960* SNP itself on the *PKDCC* dosage. Finally, using micro-computed tomography (micro-CT) coupled with dense landmark morphometric analysis of skull elements, we demonstrate the phenotypic impact of *Pkdcc* dosage on mouse craniofacial development and reveal its striking concordance with the lower jaw phenotypes and clefting predisposition associated with the *rs6740960* SNP in humans.

## Results

### Common genetic variant at 2p21 associated with lower jaw shape variation and susceptibility to clefting shows large frequency shifts in human populations

Our recent multivariate GWAS studies of normal-range facial variation in two independent European ancestry populations revealed an association of non-coding genetic variants at the 2p21 locus with lower jaw and chin shape (*Claes et al., 2018*; *White et al., 2021*). The lead SNP at this locus, *rs6740960* (A/T) (lowest meta-analysis p-value = $3.39 \times 10^{-36}$), was significantly associated with shape variation over the entire face, however, the most pronounced effects were seen in the lower face, especially in the jaw region (*Figure 1A*, *Figure 1—figure supplement 1*). Specifically, the 'T' allele (Allele Frequency = 0.497 *Claes et al., 2018*) is associated with an outward protrusion of the lower jaw and zygomatic region, coupled with a retrusion of the entire central midface. Interestingly, this same lead SNP was significantly associated with non-syndromic cleft-lip and palate (nsCL/P) in a separate GWAS from an independent cohort of European ancestry individuals (*Ludwig et al., 2017*). The risk allele for nsCL/P was the 'T' allele (*P*-value = $5.71 \times 10^{-13}$) (*Figure 1A*).

In the facial GWAS analysis, *rs6740960* is in high linkage-disequilibrium with one other genome-wide significant SNP, *rs4952552* ($r^2$=0.81 using a European cohort), located ~16 kb away (*Figure 1—figure supplement 2A*). We have prioritized *rs6740960* for functional follow-up because we observed that *rs6740960* but not *rs4952552* resides within a genomic region marked by an active enhancer chromatin signature in in vitro derived human CNCCs (*Figure 1—figure supplement 2B*). Nonetheless, we note that other SNPs in linkage with *rs6740960* may contribute to the observed genetic association of the 2p21 locus with the lower jaw shape.

We were intrigued that a common genetic variant – indeed *rs6740960* is approaching 50% allele frequency in European ancestry populations – simultaneously affects normal-range jaw shape and predisposes to clefting. To understand the prevalence of this SNP across diverse human populations, we examined allele frequencies for *rs6740960* within the 1000 Genomes Project populations (*Auton et al., 2015*). We observed a large range of 'A' vs 'T' allele frequencies, with the 'A' allele occurring at roughly 50% frequency in Europeans and South Asians (i.e. Europeans = 52%, South Asians = 58%), but at much lower frequencies in other populations (i.e. 21% in Africans and to its lowest frequency of 2% in East Asians) (*Figure 1B*). These allele frequencies were also consistent within a second human population dataset, namely the Human Genome Diversity Project (HGDP) (*Bergström et al., 2020*). Furthermore, we found that the fixation index metric, $F_{ST}$, between East-Asians and either Europeans or South-Asians was 0.48 and 0.52, respectively, indicative of high genetic differentiation between East Asians and these two populations for this specific SNP. $F_{ST}$ values range from 0 to 1 with 0 indicative

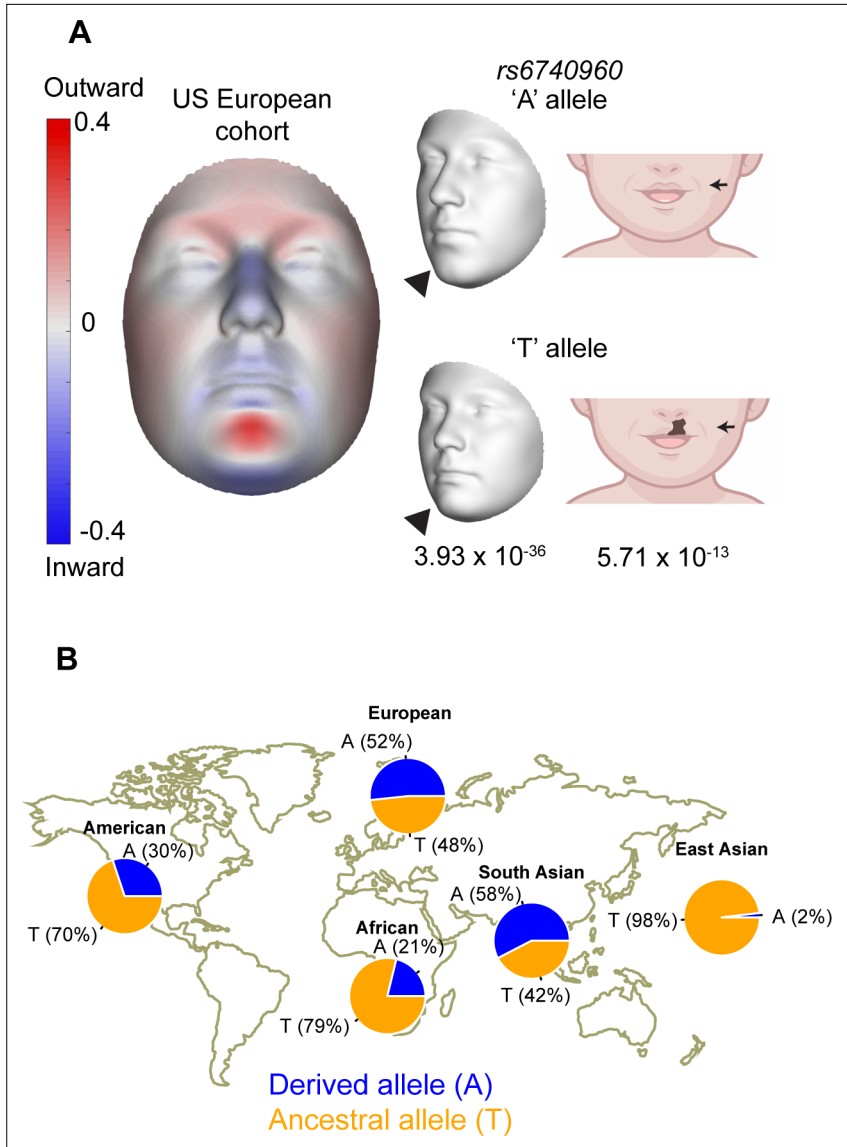

**Figure 1.** Association of *rs6740960* (A/T) with normal-range facial shape variation and its allele frequency among world populations. (**A**) Genome-wide association studies (GWAS) of normal-range facial variation in Europeans identified *rs6740960* as a lead single nucleotide polymorphism (SNP) at the 2p21 locus (*Claes et al., 2018*). 3D facial morphs and facial effects of the *rs6740960* are shown, with blue and red indicating a local shape depression and protrusion, respectively, due to the 'T' allele. Note the protrusion of the lower jaw and zygomatic regions and retrusion of the entire central midface associated with the 'T' allele. An independent GWAS linked *rs6740960* with susceptibility to non-syndromic cleft lip with or without palate in Europeans (*Ludwig et al., 2017*). (**B**) Allele frequency distribution for *rs6740960* across populations of the 1000 Genomes Project. (see also *Figure 1—figure supplements 1–2*).

The online version of this article includes the following figure supplement(s) for figure 1:

**Figure supplement 1.** Facial effects for the *rs6740960* (A/T) single nucleotide polymorphism (SNP) implicated in normal-range facial variation within the United Kingdom and the United States of America European study cohorts of *White et al., 2021*.

**Figure supplement 2.** Linkage Disequilibrium (LD) calculation, and epigenetic features, at the 2p21 locus associated with normal-range lower jaw variation (see *White et al., 2021*).

of identical allelic content whereas 1 indicates that alternate alleles are fixed in each sub-population. Large shifts in *rs6740960* allele frequency within human populations, and high $F_{ST}$ values between populations, suggest that this GWAS variant may have a distinct impact on facial shape variation and disease predisposition in different populations, with potential population-specific selection events.

## *rs6740960* resides in a craniofacial enhancer with evolutionarily modulated activity

To investigate the evolutionary history of the *rs6740960* variant, we constructed sequence alignments of extant and archaic reconstructed hominin and hominid species, centered at the SNP (*Figure 2A*). These multi-species alignments revealed that the 'T' allele (associated with a more protruding jaw morphology and predisposition to clefting) is the ancestral allele, shared with the Neanderthal and the chimpanzee, whereas the 'A' variant at this position is a derived allele unique to humans. Interestingly, the T to A substitution at this site is associated with the gain of a putative DLX binding site (*Figure 2A*), with the DLX family transcription factors playing important roles in jaw patterning and development [reviewed in *Minoux and Rijli, 2010*]. Additionally, we observed three chimp-specific mutations in the vicinity of *rs6740960,* one of which is predicted to confer gain of the ETS transcription factor family binding site (*Figure 2A*).

These observations suggest that *rs6740960* may influence facial shape by affecting the activity of a *cis*-regulatory element, and that the same element may also be modulated in a chimp-specific manner. Consistent with this hypothesis, we have previously noted that *rs6740960* resides within a putative CNCC enhancer (*Claes et al., 2018*). Furthermore, enrichment of the coactivator p300 and of the active chromatin mark H3K27ac at this region is higher in the chimpanzee CNCCs as compared to humans, suggesting that this non-coding element may have higher regulatory activity in the chimp (*Figure 2B*).

To test whether the genomic region spanning *rs6740960* is a *bona fide* craniofacial enhancer, we cloned orthologous human (with an ancestral T variant at *rs6740960*) and chimpanzee regions into a LacZ reporter vector and assayed them by conventional transgenic mouse reporter assays. The resulting activity patterns were analyzed by LacZ staining of E11.5 mouse embryos. We observed that the human element displayed specific craniofacial activity in the developing upper and lower jaw primordia (mandibular and maxillary prominences), and in the lateral nasal prominence, but had little activity in the medial nasal prominence or in other tissues of the embryo (*Figure 2C*). This pattern confirms that the genomic region overlapping *rs6740960* is a *bona fide* enhancer active during craniofacial development. For simplicity, we will hereafter refer to this element as the '*rs6740960* cognate enhancer.' The chimp ortholog of the *rs6740960* cognate enhancer was also active in the developing face, but relative to the human element, showed both stronger and broader activity, especially within the medial nasal prominence (*Figure 2C*). Although one caveat to this analysis is that the transgenic reporter was integrated randomly, and thus a subject of positional effects, we have observed weaker activity of the human enhancer across multiple embryos (see *Figure 2—figure supplement 1* for all LacZ positive embryos). Together, our data demonstrate that *rs6740960* resides within a *bona fide* craniofacial enhancer whose activity diverged in hominids.

## The *rs6740960* cognate enhancer regulates the expression of *PKDCC*

To begin probing the mechanism by which the *rs6740960* cognate enhancer can influence face shape and affect susceptibility to clefting, we first set out to identify its target gene(s). The closest annotated gene to the *rs6740960* is *C2orf91* (*LINC02898*). However, when we examined RNA-seq data from the in vitro derived CNCCs and their derivative cell types, we observed minimal if any, expression of *C2orf91* (TPM <1) (*Figure 3—figure supplement 1*), suggesting that a *bona fide* target of the *rs6740960* cognate enhancer may be a more distal gene. Analysis of cell type-specific long-range chromatin interactions can help nominate their distal target genes (*Jeng et al., 2019*; *López-Isac et al., 2019*). To examine long-range interactions of the *rs6740960* cognate enhancer during craniofacial development, we utilized an in vitro model previously developed in our laboratory, in which hESC are differentiated to CNCCs, and then further to cranial chondrocytes (*Bajpai et al., 2010*; *Rada-Iglesias et al., 2012*; *Prescott et al., 2015*; *Long et al., 2020*; *Figure 3A*). From these cell derivatives, we profiled active chromatin by H3K27ac ChIP-seq and enhancer-promoter interactions using H3K27ac HiChIP, an assay based on principles of chromatin conformation capture, which further

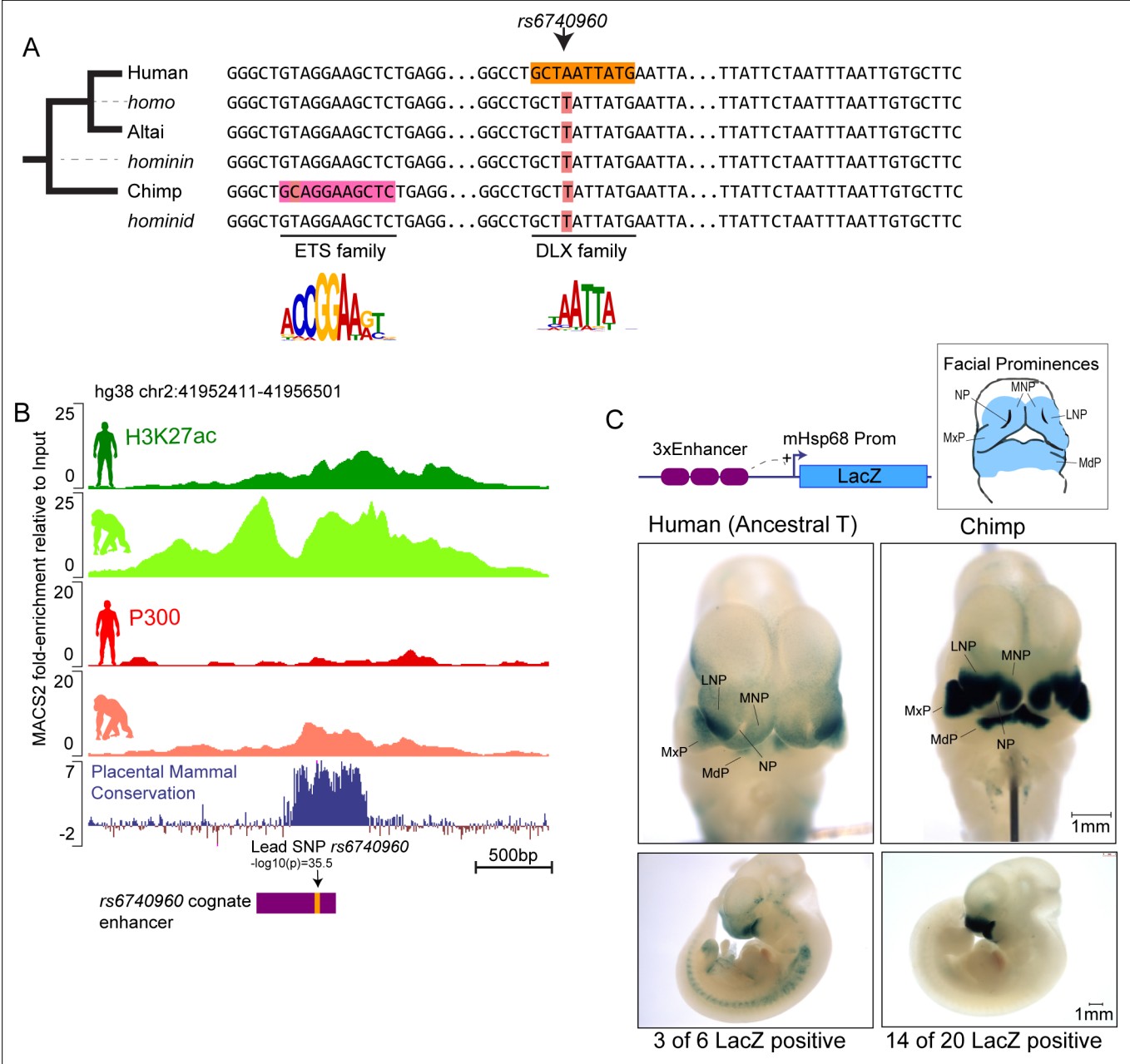

**Figure 2.** Sequence and activity changes of *rs6740960* cognate enhancer in hominids. (**A**) Multi-species abridged alignment of the genomic region surrounding *rs6740960*. Extant species are Human, Altai Neanderthal and Chimp, and ancestral reconstructed species are Homo, Hominin, and Hominid. Human-specific substitution ('A' at *rs6740960*) and chimp-specific substitution (T→C) are highlighted, along with transcription factors predicted to gain binding affinity via the substitution and their consensus sequence motifs. (**B**) Genome browser track showing the location of *rs6740960* and its overlap with H3K27ac and P300 ChIP-seq signal from human and chimpanzee CNCCs. Note higher enrichments in the chimpanzee. Region corresponding to the sequence tested in (**C**) is highlighted in purple. ChIP-seq data from *Prescott et al., 2015* (**C**) LacZ Transgenic Mouse Reporter assays performed using Human (with an ancestral 'T' allele) and Chimp orthologs of the genomic region surrounding *rs6740960*. Triplicate copies of the 500 bp sequence orthologs are used in the reporter vector. Both human and chimp orthologs show LacZ reporter activity restricted to the head and face prominences in E11.5 transgenic mice and overlapping upper and lower jaw primordia (Maxillary Prominence-MxP and Mandibular Prominence-MdP). The Chimp ortholog shows stronger activity within both MxP and MdP and within the Lateral and Medial Nasal Processes (LNP and MNP). NP – Nasal Pit. LacZ-positive animals that show the same staining profile of expression domains are numbered. See *Figure 2—figure supplement 1* for images of all LacZ-positive embryos.

The online version of this article includes the following figure supplement(s) for figure 2:

**Figure supplement 1.** LacZ transgenic mouse reporter assay results for the 500 bp Human 'T' allele and Chimp ortholog of the *rs6740960* cognate enhancer.

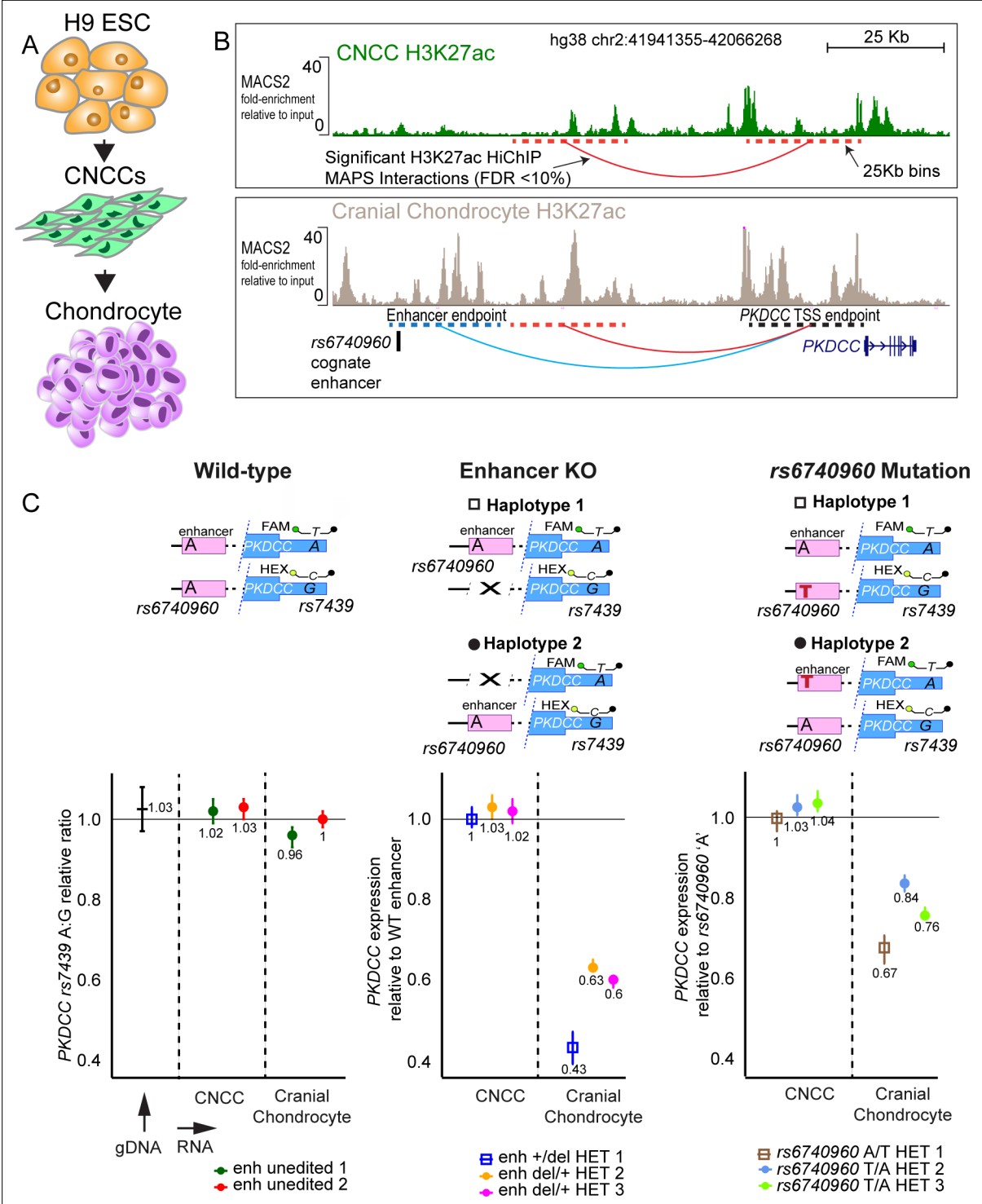

**Figure 3.** *rs6740960* regulates expression of *PKDCC* in cranial chondrocytes. (**A**) Schematic of the in vitro differentiation protocol for obtaining CNCCs and cranial chondrocytes from human embryonic stem cells (hESCs). (**B**) Mapping of long-range chromatin interactions by H3K27ac HiChIP in CNCCs and cranial chondrocytes revealed significant contacts between *rs6740960* cognate enhancer and *PKDCC* promoter in chondrocytes. H3K27ac ChIP-seq genome browser tracks at the locus from CNCCs (top) and cranial chondrocytes (bottom) are shown. Significant HiChIP interactions, called at FDR <10% and within 25 Kb bins given the resolution of this dataset (dotted lines), are shown underneath ChIP-seq tracks. Fold-change relative to input samples, as computed by the ENCODE chipseq2 pipeline, are shown on the y-axis. (**C**) Changes in allele-specific *PKDCC* expression upon deletion of *rs6740960* cognate enhancer or mutation of *rs6740960* to heterozygosity. Wild-type un-edited cells, homozygous for *rs6740960* (A/A), show no *PKDCC* allelic imbalance, as expected. Three independent CRISPR-Cas9 heterozygous enhancer deletion hESC lines show a 40–60% decrease in *PKDCC*

*Figure 3 continued on next page*

Figure 3 continued

expression for the enhancer deletion allele selectively in cranial chondrocytes. Three independent heterozygous *rs6740960* (A/T) edited lines show a 15–30% decrease in *PKDCC* expression due to the ancestral 'T' allele. FAM/HEX droplet-digital PCR (ddPCR) probes that distinguish *PKDCC* alleles based on *rs7439* (A/G) heterozygous single nucleotide polymorphism (SNP) are shown. This same SNP was used to phase the enhancer deletion or SNP mutation with *PKDCC* (see *Figure 3—figure supplements 3–4*). Genomic DNA from H9 wild-type cells was used to verify the FAM/HEX ratio of one at *rs7439*.

The online version of this article includes the following source data and figure supplement(s) for figure 3:

**Figure supplement 1.** RNA expression of lincRNA *C2orf91/LINC02898* and protein-coding gene *PKDCC,* and two marker genes of CNCC identity (*AP2α* and *NR2F1*).

**Figure supplement 2.** H3K27ac ChIP-qPCR data from H9 human embryonic stem cell (hESC) derived CNCCs and cranial chondrocyte cells, as shown in *Figure 3A and B*.

**Figure supplement 3.** Generation of a 1.2 Kb deletion at the *rs6740960* cognate enhancer using CRISPR/Cas9 genome editing of H9 human embryonic stem cells (hESCs).

**Figure supplement 3—source data 1.** Unprocessed gel blot, as SCN file, and labeled TIFF image corresponding to *Figure 3—figure supplement 3B*.

**Figure supplement 4.** Generation of A-to-T point mutation at *rs6740960* using CRISPR/Cas9 genome editing of H9 human embryonic stem cells (hESCs).

**Figure supplement 5.** *PKDCC* allele-specific droplet-digital PCR (ddPCR) expression profiling conducted from a second independent differentiation of unedited wild-type, heterozygous enhancer knock-out, and heterozygous *rs6740960* mutant human embryonic stem cell (hESC) lines.

enriches for long-range interactions between active regulatory elements via an immunoprecipitation step with an H3K27ac antibody (*Mumbach et al., 2016*). In H3K27ac ChIP-seq data, we observed that the *rs6740960* cognate enhancer is comparably enriched for H3K27ac both in CNCCs and in cranial chondrocytes, although in general, there was more H3K27ac at the locus in chondrocytes (*Figure 3B*). We further confirmed commensurate H3K27ac levels at the *rs6740960* cognate enhancer in CNCCs and chondrocytes using the more quantitative ChIP-qPCR analysis (*Figure 3—figure supplement 2*). Surprisingly, however, in the HiChIP assay, the *rs6740960* cognate enhancer makes contact with the promoter of the *PKDCC* gene selectively in chondrocytes, but not in CNCCs (*Figure 3B*). While we observed another significant contact between the *PKDCC* promoter and more proximal enhancers both in CNCCs and in chondrocytes, we detected significant contact between the *PKDCC* promoter and the *rs6740960* cognate enhancer only in chondrocytes (*Figure 3B*). *PKDCC* is an attractive target gene candidate for the *rs6740960* cognate enhancer, because it encodes a secreted tyrosine kinase involved in chondrocyte differentiation, which when mutated in mice results in skeletal defects in limbs and face, including clefting (*Imuta et al., 2009*; *Kinoshita et al., 2009*; *Bordoli et al., 2014*).

To directly test the contribution of the *rs6740960* cognate enhancer to *PKDCC* expression, we used CRISPR/Cas9 genome editing to generate hESC lines containing a heterozygous deletion of the 1.2 kB sequence corresponding to the entire, broad p300 peak spanning the *rs6740960* cognate enhancer (*Figure 3—figure supplement 3A–B*). Following differentiations of three independent heterozygous enhancer deletion lines and wild-type control hESC lines (that underwent the editing process in parallel but remained unedited) to CNCCs and cranial chondrocytes, we profiled the allele-specific expression of *PKDCC* using droplet-digital PCR (ddPCR). In ddPCR, fluorescently labeled FAM or HEX probes designed against a heterozygous SNP (*rs7439*) within *PKDCC* 3' UTR region permit allele-specific measurements of *PKDCC* expression in the presence or absence of the enhancer. The assay is internally controlled, since measurements of *PKDCC*'s expression with or without the enhancer are simultaneously assayed in thousands of individual droplets where each droplet receives a single template. By phasing the enhancer deletion strand with the corresponding *PKDCC* SNP allele (*Figure 3—figure supplement 3C*), *PKDCC*'s expression can be computed from the final number of FAM or HEX-labeled droplets using Poisson statistics. These experiments showed an approximately 40–60% decrease in *PKDCC* expression at the allele bearing the enhancer deletion. Notably, this effect was present only within the heterozygous deletion lines but absent from wild-type cells, as expected (*Figure 3C*, *Figure 3—figure supplement 5* for independent biological replicate). In concordance with H3K27ac HiChIP interactions, the effect of the enhancer deletion was present in cranial chondrocytes, but not in CNCCs. Collectively, our results demonstrate that the *rs6740960* cognate enhancer targets the *PKDCC* gene and provides a strong contribution to its expression selectively in chondrocytes.

## *rs6740960* variant causes cell-type specific allelic differences in the *PKDCC* expression

To provide direct evidence for the effect of the lead GWAS SNP itself on *PKDCC*'s expression, we used genome editing to create multiple, independent clonal *rs6740960* A/T heterozygous hESC lines (wild type H9 hESC used in our studies are A/A homozygous for *rs6740960*). Given that H9 hESC is heterozygous for other variants at the locus, we obtained cell lines in which one or the other alternate allele was mutated from A to T at *rs6740960* (*Figure 3—figure supplement 4*). Following differentiation to CNCCs and cranial chondrocytes, we compared *PKDCC* expression in these lines to the unedited cells. As expected, the effects of the single base mutation were not as large as those seen for the deletion of the entire enhancer. Nonetheless, the allele bearing the 'T' variant at *rs6740960* showed a 15–30% reduction in *PKDCC* expression across examined cell lines, and this reduction was observed only in chondrocytes (*Figure 3C*, *Figure 3—figure supplement 5* for independent biological replicate). Of note, while the 'T' mutation consistently resulted in diminished *PKDCC* expression across cell lines, the effect was larger when one allele was targeted compared to the other; similar effects were observed in heterozygous deletion lines where the enhancer has been deleted from alternate alleles (*Figure 3C*). These observations suggest that other *cis*-regulatory variants at the locus – which need not be in genetic linkage with *rs6740960* – can modulate the effects of *rs6740960* and its cognate enhancer on *PKDCC* expression. Together, our results show that the *rs6740960* variant is sufficient to affect *PKDCC* transcript levels.

## In vivo perturbation of *Pkdcc* dosage affects the shape of the jaw and palatine bones

Our molecular studies of the *rs6740960* cognate enhancer, its *cis*-mutations, and *PKDCC* expression suggest that *rs6740960* affects the normal-range facial shape and predisposition to clefting by quantitatively influencing *PKDCC* dosage during cranial chondrogenesis. To directly quantify the impact of *PKDCC* dosage changes on facial morphology, we turned to mouse models. Knockout of *Pkdcc* (also known as *Vlk*) in mice results in craniofacial malformations and clefting, indicating evolutionary conservation of *PKDCC* craniofacial function, and nominating mice as the most relevant, experimentally accessible model for directly quantifying the impact of *Pkdcc* dosage changes on facial morphology (*Imuta et al., 2009*; *Kinoshita et al., 2009*). Given that the *rs6740960* cognate enhancer affects *PKDCC* dosage by altering its expression, we perturbed *Pkdcc* gene dosage in mice through heterozygous coding mutations using previously described *Pkdcc* (*Vlk*) knockout mice (*Kinoshita et al., 2009*). The phenotypes reported in the *Pkdcc*-/- mice include perinatal lethality associated with respiratory and suckling defects, growth retardation, shortened limbs, delayed ossification, and craniofacial anomalies; at least some of these malformations have been attributed to perturbed chondrogenic differentiation and maturation (*Imuta et al., 2009*; *Kinoshita et al., 2009*). Within the cranial skeleton, *Pkdcc*-/- neonates were described to have cleft palate and shortened nasal capsule and maxilla, whereas dysmorphic craniofacial phenotypes have not been reported in heterozygous *Pkdcc*+/- mice (*Imuta et al., 2009*; *Kinoshita et al., 2009*). To confirm the reported loss-of-function phenotypes and to quantitatively characterize potential effects associated with *Pkdcc* heterozygosity, we performed high-resolution micro-computed tomography (micro-CT) imaging on wild-type, *Pkdcc*+/- and *Pkdcc*-/- mice at E18.5 (*Figure 4A*). First, we confirmed that *Pkdcc*-/- embryos (n=13) indeed displayed a shortened nasal capsule and severe clefting affecting the relevant regions of the maxilla and palatine bones as compared to wild-type embryos (*Figure 4—figure supplement 1*, *Figure 4—videos 1–2* (mandible), *Figure 4—videos 3–5* (maxilla/palatine)).

We next proceeded to examine the phenotypes of *Pkdcc*+/- mice. Given the modest, normal-range face shape changes associated with *rs6740960* in humans (*Figure 1A*), we also expected subtle phenotypes associated with *Pkdcc* heterozygosity in mice, requiring both a large animal sample size and sensitive morphometric analysis. In this vein, we utilized a pipeline that incorporated high-resolution micro-CT data from 20 wild-type and 24 *Pkdcc*+/- E18.5 mutant embryos spanning seven litters, manual bone segmentation, and dense landmarking of bone surfaces to capture subtle effects on bone shape variation (*Figure 4A*). To circumvent the limitations of sparsely labeled bone landmarks common to standard morphometric analysis, we used the software package MeshMonk, to densely landmark each specimen via a non-rigid registration of a template surface onto each specimen thereby allowing comparative bone surface evaluation (*White et al., 2019*). Using the full collection of aligned surfaces

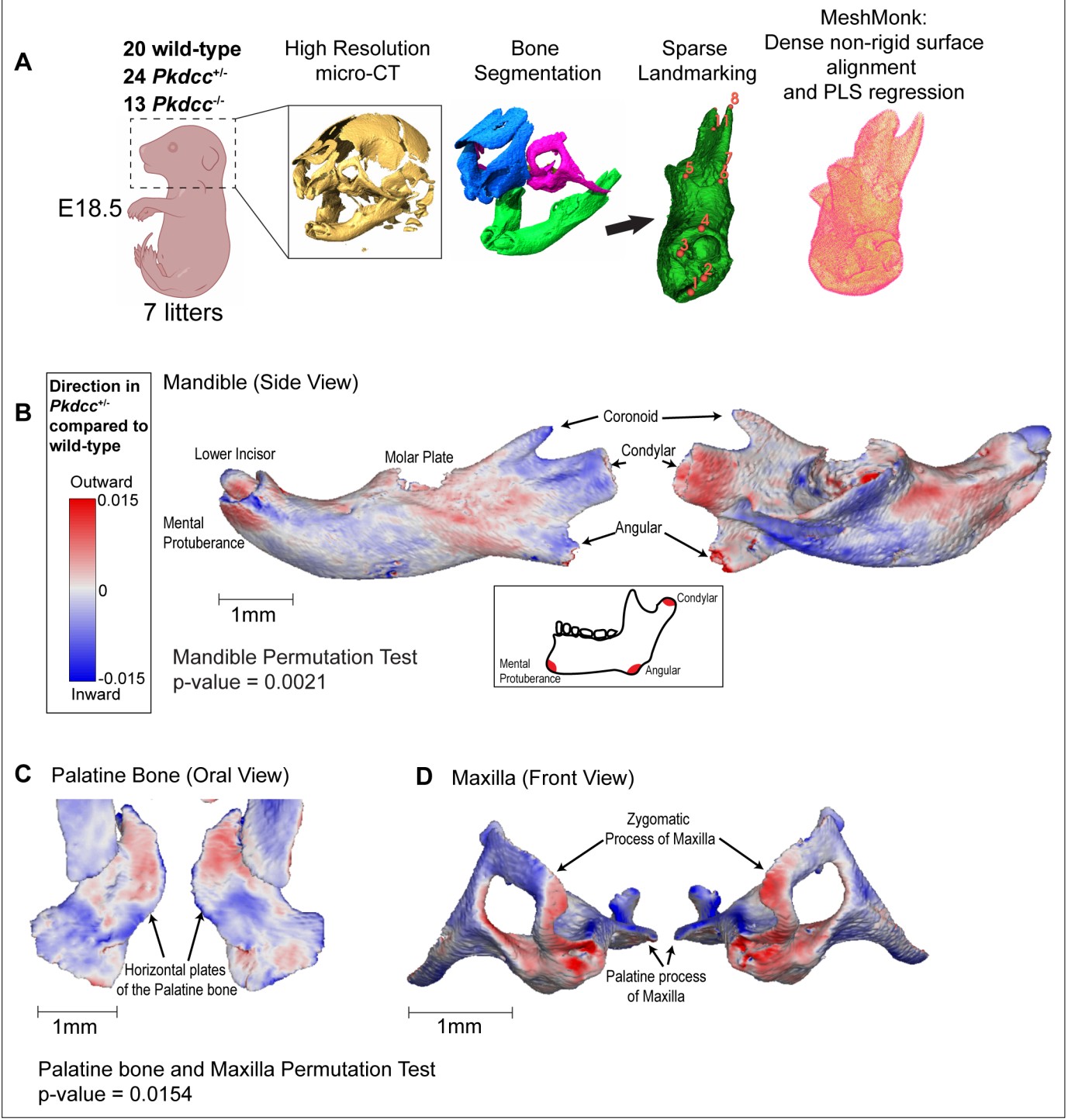

**Figure 4.** Morphogenetic assessment of *Pkdcc* dosage on cranial skeletal variation in mice. (**A**) Workflow of cranial morphometric analysis from wild-type and *Pkdcc* mutant mice. Following colony expansion, wild-type, *Pkdcc*⁺/⁻ and *Pkdcc*⁻/⁻ E18.5 embryos were fixed and imaged using high-resolution micro-CT. Following bone segmentation and sparse landmarking, bone surfaces were non-rigidly aligned to template samples with the MeshMonk software, which facilitated a dense correspondence map between all animals' surfaces. Partial least squares (PLS) regression was used to analyze genotype-to-phenotype effects. (**B, C, D**) Bone surface variation between wild-type and *Pkdcc*⁺/⁻ heterozygous mutant mice for the Mandible (**B**), Palatine Bone (**C**), and the Maxilla (**D**). Blue and red coloring represent local depression and protrusion of the bone surface, respectively, due to 50% reduction in *Pkdcc* expression. (**B**) *Pkdcc*⁺/⁻ mice exhibit overall significant shape change in the mandible (Permutation test global p-value = 0.0021). Local variation in the mandibular ramus, angular process, condylar process, and the mental protuberance contribute to the overall elongation of the mandible (See also ***Figure 4—videos 6–7***). (**C,D**) *Pkdcc*⁺/⁻ mice exhibit overall significant shape change in the Maxilla and Palatine bones (Permutation

*Figure 4 continued on next page*

*Figure 4 continued*

test global p-value = 0.015). (**C**) *Pkdcc*⁺/⁻ mice have a wider distance between the horizontal plates of the palatine bone, a structure that forms the posterior part of the hard palate (See also *Figure 4—videos 8–10*). (**D**) Significant variation in the shape of maxilla, especially within the zygomatic arches between wild-type and *Pkdcc*⁺/⁻ animals. Similar analysis performed between wild-type and *Pkdcc*⁻/⁻ animals are shown in *Figure 4—videos 1–2* (mandible), and *Figure 4—videos 3–5* (maxilla/palatine). (see also *Figure 4—figure supplement 1*).

The online version of this article includes the following video and figure supplement(s) for figure 4:

**Figure supplement 1.** Micro-CT images demonstrating cranial skeletal differences between wild-type and *Pkdcc*⁻/⁻ mutant E18.5 mouse embryos.

**Figure 4—video 1.** Medial side view of the left mandible shape effect changes between averaged wild-type and *Pkdcc*⁻/⁻ E18.5 mouse embryos. https://elifesciences.org/articles/82564/figures#fig4video1

**Figure 4—video 2.** Lateral side view of the left mandible shape effect changes between averaged wildtype and *Pkdcc*⁻/⁻ E18.5 mouse embryos. https://elifesciences.org/articles/82564/figures#fig4video2

**Figure 4—video 3.** Oral view of the maxilla and palatine bone shape effect changes between averaged wild-type and *Pkdcc*⁻/⁻ E18.5 mouse embryos. https://elifesciences.org/articles/82564/figures#fig4video3

**Figure 4—video 4.** Top view of the maxilla and palatine bone shape effect changes between averaged wild-type and *Pkdcc*⁻/⁻ E18.5 mouse embryos. https://elifesciences.org/articles/82564/figures#fig4video4

**Figure 4—video 5.** Front view of the maxilla and palatine bone shape effect changes between averaged wild-type and *Pkdcc*⁻/⁻ E18.5 mouse embryos. https://elifesciences.org/articles/82564/figures#fig4video5

**Figure 4—video 6.** Medial side view of the left mandible shape effect changes between averaged wild-type and *Pkdcc*⁺/⁻ E18.5 mouse embryos. https://elifesciences.org/articles/82564/figures#fig4video6

**Figure 4—video 7.** Lateral side view of the left mandible shape effect changes between averaged wild-type and *Pkdcc*⁺/⁻ E18.5 mouse embryos. https://elifesciences.org/articles/82564/figures#fig4video7

**Figure 4—video 8.** Oral view of the maxilla and palatine bone shape effect changes between averaged wild-type and *Pkdcc*⁺/⁻ E18.5 mouse embryos. https://elifesciences.org/articles/82564/figures#fig4video8

**Figure 4—video 9.** Top view of the maxilla and palatine bone shape effect changes between averaged wild-type and *Pkdcc*⁺/⁻ E18.5 mouse embryos. https://elifesciences.org/articles/82564/figures#fig4video9

**Figure 4—video 10.** Front view of the maxilla and palatine bone shape effect changes between averaged wild-type and *Pkdcc*⁺/⁻ E18.5 mouse embryos. https://elifesciences.org/articles/82564/figures#fig4video10

as the measured phenotype, we used partial least-squares regressions and permutation tests to compute the effect of genotype, independent of other covariates such as litter and sex, on phenotype, either at the level of complete bone shape or variation at each vertex (see **Methods**). Local regions of depression or protrusion in the shape of the bone in *Pkdcc* heterozygous mutant animals, as compared to wild-type animals, are visualized in blue and red color maps representing the regression coefficients.

Using this pipeline, we characterized the phenotypes of the mandible, maxilla, and palatine bones. We saw overall significant changes to the mandible shape, with strong local effects seen within the mandibular ramus and mental protuberance region, especially evident in the elongation of the angular process and mental region in *Pkdcc*⁺/⁻ as compared to wild-type animals, phenotypes consistent with the protruding jaw phenotype reported in GWAS (*Figure 4B*, *Figure 4—videos 6–7*). Additionally, we observed significant shape variation in the oral view of the palatine bone, especially in the distance between the symmetric horizontal plates whose eventual closure forms the base of the hard palate (*Figure 4C*, *Figure 4—videos 8–10*). A similar effect was observed, but to a lesser extent, for the palatal processes of the maxillae. The palatine and maxillary phenotypes are more subtle and distinct from the complete cleft phenotype that is present in the *Pkdcc*⁻/⁻ homozygous mice. These changes can explain why alternations in *PKDCC* dosage associated with a common genetic variant in humans increase susceptibility to clefting, but are not by themselves sufficient to cause CL/P. Finally, we examined the maxilla for shape differences between genotypes. We observed no appreciable differences within the palatine process of the maxilla, but we saw significant shape changes throughout the zygomatic processes (*Figure 4D*, *Figure 4—videos 8–10*). Interestingly, these observations may potentially explain why the *rs6740960* variant shows a modest, but significant association with cheek shape in facial GWAS (*Figure 1A*). In summary, quantitative exploration of facial phenotypes in *Pkdcc*⁺/⁻ mice has allowed us to identify specific anatomical structures that are affected by decreases in *Pkdcc* gene dosage. The remarkable concordance of these phenotypes and face shape changes associated with

the *rs6740960* substitution in humans provides strong support to our hypothesis that this genetic variant affects normal-range and disease-associated facial variation through regulating the expression of *PKDCC*.

## Discussion

Our study provides an in-depth functional analysis of a non-coding genetic variant associated with both normal-range and disease-associated variation in a human morphological trait. Based on our collective results we propose a mechanism of *rs6740960* function. The *rs6740960* SNP resides in and modulates the activity of a craniofacial enhancer active in the developing jaw primordia. This enhancer regulates the expression of *PKDCC* in cranial chondrocytes. The *PKDCC* gene encodes a secreted tyrosine kinase, which remodels the extracellular matrix during cartilage development and affects the timing of proliferative chondrocyte differentiation and subsequent bone formation (*Bordoli et al., 2014*). Although in mice *Pkdcc/Vlk* is also important for long bone development in limbs (*Imuta et al., 2009*; *Kinoshita et al., 2009*), the craniofacial activity of the cognate *rs6740960* enhancer restricts the SNP effects on *PKDCC* dosage to the developing face. Thus, a parsimonious explanation for the association of *rs6740960* with facial variation is through the sensitivity of cranial chondrogenesis, and in turn facial skeletal development, to the *PKDCC* dosage. This may be especially pronounced during lower jaw development, where the transient Meckel's cartilage impacts mandibular form (*Svandova et al., 2020*), and mandibular chondrocytes can directly transform into bone cells (*Jing et al., 2015*). Indeed, we demonstrated that in mice 50% reduction in *Pkdcc* dosage results in quantitative changes in the shape of several mandibular regions.

Our observations can also explain why *rs6740960* is associated with nsCL/P. The reduction in *Pkdcc* dosage increased the distance between the horizontal plates of the palatine bones (and the palatal plates of the maxillae to a lesser extent) which ultimately need to fuse to form the secondary palate, but by itself, is not sufficient for clefting. This makes sense, given that the frequency of the risk-conferring *rs6740960* 'T' allele in the European ancestry population (even in a homozygous setting) is two orders of magnitude higher than the frequency of orofacial clefting in the same population. Thus, by definition, the risk variant must work additively or synergistically with other variants acting on genes affecting palate development, either at the same locus or at different loci, and/or with environmental factors. These include prenatal exposure to alcohol, nicotine, viruses, maternal diabetes, and methyl donor deficiency, all of which are linked to an increased prevalence of orofacial clefts (*Martinelli et al., 2020*). Interestingly, unaffected relatives of individuals with nsCL/P show distinctive facial features, which have been quantitatively characterized to define a subclinical 'endophenotype' that may reflect a heightened susceptibility to clefting (*Indencleef et al., 2021*; *Weinberg, 2022*). Subsequent GWAS analysis revealed loci associated with nsCL/P endophenotype in a healthy, unselected population (*Indencleef et al., 2021*). Notably, one of the genome-wide significant lead SNPs from this analysis, *rs4952552*, maps to the *PKDCC* locus, suggesting that the impact of *rs6740960* on clefting risk is likely modulated by the presence of other common variants acting on *PKDCC*.

It is not entirely clear what the embryological origins of the maxillary and palatine changes observed in *Pkdcc* heterozygous mutants, particularly given the intramembranous origin of these bones. One possibility is that these changes are a secondary consequence of effects taking place in the adjacent endochondral cranial base. Another possibility worth future exploration is that in addition to chondrocytes, mesenchymal tissues forming the bony structures of the palate are also sensitive to *Pkdcc* dosage. Consistent with this possibility, *Kinoshita et al., 2009* showed that *Pkdcc* is expressed in the CNCC-derived mesenchyme populating the palatal shelves. Regardless, the quantitative phenotypes we describe in *Pkdcc* heterozygous mice are consistent with an increased susceptibility of clefting in Europeans carrying the risk-conferring *rs6740960* allele.

The activity of the *rs6740960* cognate enhancer diverged in hominids, suggesting that this element may have contributed to the divergence in craniofacial morphology between hominid species. Our analysis also revealed drastic variation in *rs6740960* allele frequency across modern human populations, with Europeans and South Asians showing high heterozygosity for this SNP, and East Asians exhibiting prevalence of the ancestral 'T' allele. What could be the potential adaptive advantages for the increased frequency of the human-derived allele in European and South Asian populations? Our analysis suggests that the variation in jaw phenotype associated with the *rs6740960* was partly driven by variation in the angular process of the mandible. The angular process, or mandibular angle,

attaches to the masseter muscle which generates the majority of bite-force during mastication (*Sella-Tunis et al., 2018*). Could the derived 'A' allele facilitate masticatory adaptations influenced by dietary changes along population boundaries? Was the increase in the derived allele frequency associated with the elevated susceptibility to clefting associated with the ancestral allele in the European population? And if so, are East Asians buffered from these effects by other, compensatory genetic variants? Or, alternatively, could the advantageous benefit be purely an aesthetic one, with a less or more protruding jaw appearance affecting sexual selection differently in distinct human populations and cultures? While we do not have answers to these questions at present, our results merit further consideration in the context of human facial adaptations and evolution.

## Methods

### Cell culture and differentiation of hESCs to CNCCs and cranial chondrocytes

Human Embryonic Stem Cell (hESC) H9 line (directly obtained from WiCell Research Institute [catalog number WA09]) was cultured and differentiation to CNCCs using a protocol previously described in *Prescott et al., 2015*. Briefly, the hESC H9 line was authenticated by next-generation sequencing experiments and was verified to be free of mycoplasma contamination. The hESC H9 cell line is excluded from the catalog of commonly misidentified cell lines curated by the International Cell Line Authentication Committee. hESCs were plated onto tissue-culture six-well plates pre-coated with Matrigel (Corning 356231) using mTESR-1 medium (StemCell Technologies 85850) and grown for 5–8 days with daily medium replacement. Cells were passaged 1:6 using RELESR (StemCell Technologies 05872) for detachment of hESCs. For differentiation of hESCs to CNCCs, large hESC colonies were detached using 2 mg/mL Collagenase IV (Fisher 17104019) resuspended in KnockOut DMEM (Gibco 10829018), and incubation for 30 min – 1 hr at 37 °C. Intact cell clumps were washed in PBS and plated into Neural Crest Differentiation medium (NDM) in 10 cm Petri Dishes incubated at 37 °C. Media was changed daily until Day 4, changing plates every day. On Day 4, neuroectoderm spheres were left undisturbed until Day 7 to promote attachment to the dish. From Day 7–11 CNCCs emerged from attached neural spheres, and NDM media was changed daily without disturbing attached spheres and CNCCs. To transition CNCCs to passage 1, neural spheres were removed by gentle aspiration, the plate was washed with PBS, and the remaining CNCCs were detached using 50% Accutase diluted in PBS. After incubation at 37 °C for 1 min, CNCC were disaggregated with gentle pipetting. CNCCs were plated onto fibronectin-coated six-well tissue-culture treated plates with Neural Crest Maintenance media (NMM), left to be attached for 15 min at 37 °C, and medium containing Accutase was replaced with fresh pre-warmed NMM. Medium was replaced daily without tilting plates. Cells were split 1:3 every 2 days, using 50% accutase, and the same procedure of allowing CNCCs to attach to the plate for 15 min at 37 °C, and immediately replacing accutase containing medium with fresh pre-warmed NMM. One day after splitting to passage 3, cells were transitioned to NMM + BMP2 and ChIRON medium (NMM + BC).

Passage 4 CNCCs were harvested for experiments, or differentiated further to cranial chondrocytes. For cranial chondrocytes differentiation, NMM + BC medium was replaced with Chondrocyte Differentiation Medium (CDM) without TGFβ3 supplement, with an intermediate PBS wash between media change (day 0). Medium was changed on day 1 and day 4 with CDM supplemented with TGFβ3, and cells were harvested on day 5. To disassociate single-cell chondrocytes from the extracellular matrix, we used the protocol described in *Makki et al., 2017*. Briefly, cells were washed with PBS, and incubated with digestion medium (1 mg/mL Pronase (Roche 10165921001), 1 mg/mL Collagenase B (Roche 11088815001), 4 U/mL Hyaluronidase (Sigma H3506), resuspended in KnockOut DMEM) for 1 hr at 37 °C, with agitation every 15 min. Digestion medium was aspirated after centrifugation (5 min, 100 r.c.f), and the cell pellet was washed twice with PBS, with centrifugation between each wash.

Neural Crest Differentiation Medium (NDM): 1:1 Neurobasal Medium (Thermo Scientific 21103049) and DMEM F-12 medium (GE Healthcare SH30271.01), 0.5 X Gem21 NeuroPlex supplement (B-27) (Gemini 400–160), 0.5 X N2 NeuroPlex supplement (Gemini 400–163), 20 ng/mL bFGF (PeproTech 100-18B), 20 ng/mL EGF (PeproTech AF-100–15), 5 ug/mL bovine insulin (Gemini 700–112 P), 1 X Glutamax supplement (Thermo Fisher 35050061), 1 X Antibiotic-Antimycotic (Gibco 15240062).

Neural Crest Maintenance Medium (NMM): 1:1 Neurobasal Medium (Thermo Fisher 21103049) and DMEM F-12 medium (GE Healthcare SH30271.01), 1 mg/mL BSA (Gemini 700–104 P), 0.5 X Gem21 NeuroPlex supplement (B-27) (Gemini 400–160), 0.5X N2 Neuroplex supplement (Gemini 400–163), 1 mg/mL BSA (Gemini 700–104 P), 20 ng/mL bFGF (PeproTech 100-18B), 20 ng/mL EGF (Pepro-Tech AF-100–15), 1 X Glutamax supplement (Thermo Scientific 35050061), 1 X Antibiotic-Antimycotic (Gibco 15240062).

NMM +BMP2/ChIRON: NMM medium supplemented with 50 pg/mL BMP2 (PeproTech 120–02) and 3 uM CHIR-99021 (Selleck Chemicals S2924).

CDM: DMEM High Glucose Medium (Cytiva SH30243.01), 5% FBS, 1 X ITS+ Premix (Corning 354352), 1 X Sodium Pyruvate (Gibco 11360070), 50 ug/mL L-Ascorbic Acid (Sigma-Aldrich A4403), 0.1 uM Dexamethasone (Alfa Aesar A17590), 10 ng/mL TGFβ3 (PeproTech 10036E), 1 X Antibiotic-Antimycotic (Gibco 15240062).

## H3K27ac HiChIP and ChIP

CNCCs and Chondrocytes were detached as singled cells and resuspended in BSA and Serum-free NMM or CDM, respectively (1 million cells/mL), fixed in 1% formaldehyde with end-over-end rotation at room temperature for 10 min, quenched with 0.2 M glycine solution for 5 min, and pelleted by centrifugation (5 min, 1350 r.c.f, 4 °C). Cell pellet was washed twice with cold PBS, with centrifugation between washes, and the cell pellet was flash-frozen in liquid nitrogen and stored at –80 °C.

HiChIP experiments were conducted following the protocol described in *Mumbach et al., 2016* using MboI digestion, and ChIP experiments were performed as described in *Long et al., 2020*. In both experiments, we used Active Motif Rabbit Polyclonal H3K27ac ChIP-grade antibody (Cat No 39133). HiChIP experiments were performed on eight biological replicates of CNCC and chondrocyte differentiations, and both HiChIP and 1% Input libraries were sequenced on the Illumina NovoSeq 6000 (2 × 100 bp) platform resulting in 244 million non-duplicate reads. HiChIP data was processed using HiC-Pro pipeline with 25 Kb resolution (*Servant et al., 2015*), the recommended resolution of this data prescribed by HiC-Res (*Marchal et al., 2020*). HiC-Pro was used as a preprocessing workflow to align reads to the hg38 genome, and to save only valid read pairs (i.e. from *cis* long and short-range interactions, and intra-chromosomal interactions). Significant 3D interactions (10% False Discovery Rate) were called using MAPS (*Juric et al., 2019*).

To call significant 3D interactions, MAPS requires precomputed 1-dimensional (1D) ChIP peaks, ideally called from independent ChIP-seq experiments. For this purpose, H3K27ac ChIP-seq peaks for CNCCs were identified using published Human H3K27ac ChIP-seq data from *Prescott et al., 2015*. To call ChIP-seq peaks in cranial chondrocytes, we performed H3K27ac ChIP-seq experiments from three additional chondrocyte differentiations. ChIP-seq data were aligned to the hg38 genome using a BWA MEM aligner (*Li and Durbin, 2009*), and broad peaks were identified using macs3 (*Zhang et al., 2008*). Finally, for visualizing comparable H3K27ac ChIP signal for CNCC and chondrocytes in *Figure 3B*, we performed H3K27ac ChIP-seq on a single hESC → CNCC → chondrocyte differentiation. Signal tracks showing H3K27ac or P300 ChIP fold-enrichment relative to the input samples were produced from the ENCODE Transcription Factor and Histone ChIP-Seq processing pipeline (RRID:SCR_021323; version 2.2.2) (https://github.com/ENCODE-DCC/chip-seq-pipeline2; *ENCODE-DCC, 2024a*).

## LacZ transgenic mouse experiments

Enhancer elements were cloned in triplicate copy into a LacZ reporter plasmid immediately upstream of an Hsp68 promoter and LacZ-P2A-tdTomato fusion gene, all surrounded by core insulator elements. Orthologs of the *rs6740960* cognate enhancer from Human and Chimp (each 500 bp in length) were amplified from genomic DNA extracted from Human H9 and Chimp C0818 cell lines. H9 is homozygous for the 'A' (derived) allele of *rs6740960*. To create a human homolog bearing the 'T' ancestral allele, we utilized PCR primer bearing this allele and PCR fragment ligation to recreate a mutagenized sequence. Mouse transgenesis experiments, conducted by Cyagen, were performed as described in *Samuels et al., 2020*. Briefly, reporter plasmids were linearized, injected into fertilized mouse oocytes, where they randomly integrated into the genome, oocytes were implanted into recipient females, and allowed to develop to the E11.5 embryonic stage. Embryos were harvested, genotyped for positive reporter integration, and stained with X-gal to reveal the anatomical structures where the reporter transgene was expressed. We required at least three LacZ-positive embryos with consistent

LacZ staining profiles for expression domains to be considered reproducible. Multiple injection experiments were needed to meet this requirement.

## Generation of enhancer knock-out cell lines using CRISPR/Cas9

Deletion of the *rs6740960* cognate enhancer and A-to-T replacement of the lead SNP in H9 hESCs was accomplished with CRISPR/Cas9 genome editing using the CRISPR RNA 'GTGGGGATTGCG CTAACTCA' synthesized by IDT and homology-direct repair (HDR) template bearing either the 1.2 Kb enhancer deletion, or the wild-type enhancer bearing the alternate 'T' allele of *rs6740960*. HDR templates were cloned independently into an AAV2 production plasmid (Addgene 32395), using left and right homology arms cloned from H9 genomic DNA (Left Arm Primers GTGCAGAGCTGTCTG / ATGTTTTACTAGCTCCTGCTAT; Right Arm Primers AGGGAGGATGGGAAGGA / GACAGGCAAGAG ACTGACATA). AAV HDR donor plasmid, and Capsid 2 and AD5 helper plasmids, were co-transfected into HEK-293FT cells (Invitrogen R70007), and after 48 hr, high-titer virions were purified by Iodixanol ultracentrifugation using a modified protocol provided in *Martin et al., 2019*.

H9 hESCs were cultured in mTESR-1 medium supplemented with 10 uM ROCKi (Y27632; StemCell Technology 72304) for 24 hr, washed with PBS, and brought to single-cell pellet with Accutase treatment for 5 min at 37 °C and gentle centrifugation (100 r.c.f; 5 min). 1 million hESCs were resuspended with 100 uL Lonza Amaxa P3 nucleofection solution (Lonza V4XP-3024), the annealed, pre-complexed crRNA:tracrRNA guide-RNA duplex (3 uL of 50 uM gRNA duplex), and 19 ug Alt-R S.p. HiFi Cas9 v3 Nuclease (IDT 1081060) (2.5:1 Cas9:gRNA molar ratio). Cells were electroporated using the CA-137 setting on the Lonza 4D Amaxa Nucleofector, immediately plated onto precoated matrigel plates with 100 K MOI AAV2 HDR, and cultured with mTESR1 + 10 uM ROCKi for 96 hr. Edited single-cells were passaged sparsely (600 cells per well of six-well plate), and allowed to grow for 7 days. Clonal expanded colonies were manually picked, further expanded, and genomic DNA was isolated using a Quick-Extract Buffer (Lucigen GE09050). Successful heterozygous deletion lines were identified using the PCR primers CCCCCACCCATCAGTCATTC and GGTTGTGCCTCATAGTGCCT and Q5 Polymerase (NEB M0491S). Edited lines showed a 2.4 kB band corresponding to the deletion allele in contrast to the 3.6 kB unedited, wild-type band. Heterozygous *rs6740960* A/T lines were validated using droplet-digital probes and primers against *rs6740960* (see below). Successful A-to-T heterozygous replacement clonal lines were identified if they displayed a 50:50 FAM:HEX positive droplet ratio in their genomic DNA. Clonal lines that underwent the same editing procedure, but emerged un-edited were chosen as wild-type controls for all expression assays. To phase the enhancer deletion allele with *PKDCC* heterozygous SNP *rs7439*, we used the PCR primers CAGCAGCCAGAG ATCCTTTAG and AAGGAAAGCCTCCACTTCTTT to amplify a 12.7 kB region encompassing the enhancer and *rs10168129* and *rs72792594*, the two closest pre-phased SNPs obtained from H9 10 X whole-genome sequencing data (*Long et al., 2020*). PCR products were sent for long-read Oxford Nanopore sequencing, and phasing was conducted by manual examination of allele co-occurrence in the returned long reads.

## Droplet digital experiments for *rs6740960* H3K27ac signal and PKDCC expression

Droplet digital PCR (ddPCR) experiments were performed using protocols for the ddPCR Supermix for Probes (No dUTP) (Bio-Rad 1863025), droplet generator, and droplet reader provided by BioRad. To assay allele-specific expression of *PKDCC*, *rs6740960* cognate enhancer heterozygous deletion H9 lines, *rs6740960* (A/T) heterozygous H9 lines, and un-edited wild-type H9 cells were differentiated to CNCCs and chondrocytes. Total RNA was extracted using TRIzol Reagent according to manufacturer's protocol (Thermo Scientific 15596026), and 2 ug of total RNA was DNase I treated and reverse-transcribed to cDNA using random Hexamer oligos according to Thermo SuperScript IV VILO master mix kit (Thermo Scientific 11766050). cDNA was diluted 40 X for use as input material in RT-ddPCR experiments.

IDT Locked Nucleic Acid (LNA) FAM and HEX fluorescently labeled probes were designed to distinguish the 'A' and 'T' alleles of *rs6740960* in *rs6740960* (A/T) heterozygous H9 lines (56-FAM/ATT+CATA-A+T+T+AGCA+GGCCATG/3IABkFQ/ and /5HEX/ATT+CATAA+T+A+AGCA+GGCCATG/3IABkFQ/). Forward and reverse PCR primers to amplify the 119 bp region surrounding *rs6740960* were CTTT CTTCTGTGAGCTGTAGCA and TCCACGCTGGGTTTAGTTATTT. Similarly, probes were designed to

assay the allele-specific expression of *PKDCC* in *rs6740960* cognate enhancer heterozygous deletion H9 lines and *rs6740960* (A/T) heterozygous H9 lines. Specifically, FAM/HEX probes designed against heterozygous SNP *rs7439* (A/G) located within *PKDCC*'s 3'UTR (/56-FAM/CGG+G+A+G+A+TG-C/3IABkFQ/ and /5HEX/CGG+G+G+GATGC/3IABkFQ/). This 88 bp 3'UTR region was amplified with TCGTGGAGTGTTCTCTCA and AGCCTCCCCAGGTAC PCR primers. To validate that fluorescently labeled probes designed against heterozygous SNPs were unbiased in their binding affinities, H1 hESC genomic DNA was used as a quality control to ensure a 50:50 FAM:HEX ratio for the *rs6740960* SNP (H1 hESCs are naturally heterozygous for *rs6740960*), and H9 hESC genomic DNA was used as a quality control to a ensure a 50:50 FAM:HEX ratio for the *rs7439* SNP.

BioRad QuantSoft Analysis Pro v1.0 software was used to analyze ddPCR experiments. QuantSoft uses Poisson statistics to estimate the number of FAM-positive and HEX-positive droplets in the starting reaction. These ratios were plotted in R to quantify allele-specific *PKDCC* expression and to identify *rs6740960* (A/T) heterozygous edited cell lines.

## Linkage disequilibrium and imputation analysis of *rs6740960*

The SNP *rs6740960* was imputed in our previous GWAS with a high imputation INFO score of 0.91 (*Claes et al., 2018*; *White et al., 2021*). Imputation and LD calculation were based upon the reference haplotypes from the pre-phased genotypes of European individuals within the 1000 Genomes Project reference panel phase 3. Imputation INFO score values range from 0 to 1 with 0 meaning complete lack of support for a SNP's allelic phase, and 1 representing SNPs that were genotyped. Higher values indicate higher probabilities of a SNP's phase based on haplotype configurations within the reference panel. The SNP *rs4952552* was directly genotyped in our previous GWAS study and was found to be in high LD with *rs6740960* ($r^2$=0.81 using a European cohort), which contributes to the high imputation score of the latter variant.

## RNA-seq and ATAC-seq data and analysis

RNA-seq datasets were obtained from previously published studies. hESC data was obtained for H9 cells from *Prescott et al., 2015*, CNCC data were obtained from both (*Prescott et al., 2015*), and (*Yankee et al., 2023*), which reflect unique in vitro hESC to CNCC differentiation protocols. Cranial chondrocyte data were obtained from *Long et al., 2020*. All samples were uniformly processed using the ENCODE long read RNA-seq pipeline (RRID:SCR_025093; version 2.1.0) (https://github.com/ENCODE-DCC/long-read-rna-pipeline; *ENCODE-DCC, 2023*), which uses the STAR aligner for read mapping, and RSEM software for Transcripts Per Million (TPM) calculation.

ATAC-seq dataset for human CNCCs (H9 derived) were obtained from *Prescott et al., 2015*. ATAC-seq datum was processed using the ENCODE ATAC-seq pipeline (RRID:SCR_023100; version 2.2.3) (https://github.com/ENCODE-DCC/atac-seq-pipeline; *ENCODE-DCC, 2024b*).

## Micro-CT of E18.5 *Pkdcc* mutant mice

*Pkdcc* knock-out mice were created by *Kinoshita et al., 2009* by replacing the first exon with a Venus - SV40 polyA tag. Cryogenic preserved two-cell oocytes were obtained from RIKEN BioResource Research Center (RBRC057275) and rederived at Stanford University's Transgenic, Knockout, and Tumor Model Center (TKTC). Founder animals were genotyped using a triplicate primer set (ggca tggacgagctgtacaa, GGCTCAGGCTACACTAAGGC, CTGGGTTCTGCACCTAGCTG), that resulted in a single 332 bp amplicon for wild-type animals, a single 594 bp amplicon for homozygous-tagged animals, and dual bands for heterozygous animals. Mice were housed at the Animal Veterinary Center at Stanford University. All procedures involving animal care and use were conducted under a pre-approved protocol by the Administrative Panel on Laboratory Animals Care at Stanford University (APLAC Protocol No. 30364).

Mouse embryos were collected at E18.5 developmental stage, and fixed in 4% Paraformaldehyde for 7 days at 4 °C. Embryos were imaged on Bruker Skyscan 1276 micro-CT imager using the following settings (15 um pixel size, 2 K resolution, 0.25 mm Al filter, ~750 ms exposure, 360° scanning, rotation step 0.5 degree, 100% partial width, 2 frame averaging). Bruker NRecon software was used for 3D reconstruction using the settings smoothing = 2, ring artifacts correction = 20, beam hardening = 0%, Gaussian Smoothening Kernel. Volume-of-interest (VOI) were isolated for the head region of each animal using Bruker DataViewer software. Mandible, maxilla, and palatine

bones were segmented and roughly landmarked for each VOI using Amira 3D 2021.1 software. We used landmarks recommended by *Ho et al., 2015*. Segmented bone surfaces were exported as Wavefront OBJ format and landmarks were exported as fiducial CSV formats for morphometry analysis.

## Morphometry analysis for E18.5 embryos

Morphometric differences in cranial bone surfaces of wild-type, *Pkdcc*$^{+/-}$, and *Pkdcc*$^{-/-}$ E18.5 embryos were analyzed in Matlab 2021b. The MeshMonk toolbox (*White et al., 2019*) was developed for spatially dense phenotyping of 3D surface meshes of anatomical structures and has been used extensively in studies of craniofacial variation in humans (*Claes et al., 2018*; *Liu et al., 2021*; *White et al., 2021*). MeshMonk performs a non-rigid registration of a template mesh onto each target mesh. Thereafter each target sample is represented in terms of the topology of the template mesh. The vertices of each registered surface are spatially-dense analogs of traditional anthropometric landmarks. In this study, the mandible, maxilla, and palatine bone surfaces from one representative wild-type animal was chosen as the template sample, and the corresponding structures in each sample were registered with this template.

Further statistical analysis was performed using in-house software developed by authors HM and PC in MATLAB 2021b. All landmark configurations were aligned to their mean and scaled to unit size by generalized Procrustes analysis. Dimensionality of landmark variation was reduced by projection onto the principal components (PCs) explaining 96% of the variation. The partial effect of genotype (independent of litter and sex) on the shape was tested with partial least-squares regressions and permutation test (10,000 permutations) on the partial effect size (partial R$^2$) as described in *Shrimpton et al., 2014*. The categorical covariates sex and litter were each coded as k-1 binary 'dummy' variables, where k is the number of categories. To test for effects at the level of the whole structure, principal component scores were normalized to have unit variance along each PC, and the partial effect size is the variance explained in these normalized PCs. To test for effects at each vertex the principal component projection was reversed and partial R$^2$ was computed for each vertex. The regression coefficients represent the change in the shape of the bone between wild-type and *Pkdcc* mutant animals are visualized as color maps showing change in the direction perpendicular to the surface at each vertex (blue = inward; red = outward).

## Materials availability

Edited human ESC cell lines (H9) and other research reagents generated in this study are available to academic researchers by contacting the corresponding author.

## Acknowledgements

We thank members of the Wysocka lab for helpful discussions, comments, and advice. We thank Naz Koska for help with mouse husbandry, Larissa Sambel for help with cell culture and ddPCR experiments, Seppe Goovaerts for help with SNP imputation, and Drs. Heather Szabo Rogers and Sahin Naqvi for comments on the manuscript. *Pkdcc/Vlk* targeted *Mus musculus* strain (No. RBRC057275), originating from *Kinoshita et al., 2009*, was obtained from RIKEN BioResource Research Center (Acc No. CDB0434K and website http://www2.clst.riken.jp/arg/mutant%20mice%20list.html) This research was supported by the NIH grant R01 DE027023 and funding from the Howard Hughes Medical Institute. JW was supported by a Lorry Lokey endowed professorship, the NOMIS foundation award and a Stinehart Reed award. JM was supported by an NSF Postdoctoral Research Fellowship in Biology (Award No. 1711847).

## Additional information

### Competing interests

Joanna Wysocka: J.W. is a paid member of Camp4 and Paratus biosciences scientific advisory boards. The other authors declare that no competing interests exist.

## Funding

| Funder | Grant reference number | Author |
|---|---|---|
| National Science Foundation | 1711847 | Jaaved Mohammed |
| National Institute of Dental and Craniofacial Research | DE027023 | Susan Walsh<br>John R Shaffer<br>Seth M Weinberg<br>Peter Claes<br>Joanna Wysocka |
| Howard Hughes Medical Institute | | Joanna Wysocka |
| NOMIS Stiftung | | Joanna Wysocka |

The funders had no role in study design, data collection and interpretation, or the decision to submit the work for publication.

## Author contributions

Jaaved Mohammed, Resources, Data curation, Software, Formal analysis, Funding acquisition, Validation, Visualization, Methodology, Writing – original draft, Project administration; Neha Arora, Karissa Hansen, Maram Bader, Data curation, Formal analysis; Harold S Matthews, Software, Formal analysis, Investigation, Methodology; Susan Walsh, Methodology; John R Shaffer, Seth M Weinberg, Investigation; Tomek Swigut, Conceptualization, Resources, Data curation, Software, Formal analysis, Supervision, Validation, Investigation, Visualization, Methodology, Project administration; Peter Claes, Software, Investigation, Methodology; Licia Selleri, Formal analysis, Investigation, Methodology; Joanna Wysocka, Conceptualization, Resources, Supervision, Funding acquisition, Investigation, Methodology, Writing – original draft, Project administration, Writing – review and editing

## Author ORCIDs

Jaaved Mohammed (iD) http://orcid.org/0000-0002-7053-5575
John R Shaffer (iD) http://orcid.org/0000-0003-1897-1131
Tomek Swigut (iD) http://orcid.org/0000-0002-7649-6781
Joanna Wysocka (iD) https://orcid.org/0000-0002-6909-6544

## Ethics

This study was performed in strict accordance with the recommendations in the Guide for the Care and Use of Laboratory Animals of the National Institutes of Health. Mice were housed in RAFII facility at Stanford University, with free access to food and water. All procedures involving animal care and use were conducted under a pre-approved protocol by the Administrative Panel on Laboratory Animals Care at Stanford University (Protocol ID 30364).

## Decision letter and Author response

Decision letter https://doi.org/10.7554/eLife.82564.sa1
Author response https://doi.org/10.7554/eLife.82564.sa2

---

# Additional files

## Supplementary files
• MDAR checklist

## Data availability
Sequencing data generated in this study are available from the Gene Expression Omnibus (GEO Accession Number GSE212234).

The following dataset was generated:

| Author(s) | Year | Dataset title | Dataset URL | Database and Identifier |
|---|---|---|---|---|
| Mohammed J, Arora N, Matthews HS, Hansen K, Bader M, Shaffer JR, Weinberg SM, Swigut T, Claes P, Selleri L, Wysocka J | 2023 | A common cis-regulatory variant impacts normal-range and disease-associated human facial shape through regulation of PKDCC during chondrogenesis | https://www.ncbi.nlm.nih.gov/geo/query/acc.cgi?acc=GSE212234 | NCBI Gene Expression Omnibus, GSE212234 |

The following previously published datasets were used:

| Author(s) | Year | Dataset title | Dataset URL | Database and Identifier |
|---|---|---|---|---|
| Long HK, Osterwalder M, Welsh IC, Hansen K, Davies JOJ | 2020 | Phasing H9 human embryonic stem cell line | https://www.ncbi.nlm.nih.gov/bioproject/PRJNA648128 | NCBI BioProject, PRJNA648128 |
| Prescott SL, Srinivasan R, Marchetto MC, Grishina I, Narvaiza I, Selleri L, Gage FH, Swigut T, Wysocka J | 2015 | Enhancer divergence and cis-regulatory evolution in the human and chimpanzee neural crest | https://www.ncbi.nlm.nih.gov/geo/query/acc.cgi?acc=GSE70751 | NCBI Gene Expression Omnibus, GSE70751 |
| Yankee TN, Oh S, Winchester EW, Wilderman A, Robinson K, Gordon T, Rosenfeld JA, VanOudenhove J, Scott DA, Leslie EJ, Cotney J | 2023 | Transcriptomic Atlas of Embryonic Human Craniofacial Development | https://www.ncbi.nlm.nih.gov/geo/query/acc.cgi?&acc=GSE197513 | NCBI Gene Expression Omnibus, GSE197513 |

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
