## [Editor Report]

This provides an important exemplar of a coordinated body of work (using differentiated human induced pluripotent stem cells [iPSCs] and phenotyping of a mouse mutant) to dissect the mechanism by which a candidate human single nucleotide variant (SNV) may influence both shape variation in the oro-mandibular region, and susceptibility to cleft lip/palate. The extensive use of genome editing to introduce both targeted deletions and single nucleotide variants, taking into account that each allele has a different natural haplotype background associated with a differing functional readout, is especially compelling. As well as being of general interest to craniofacial biologists, the iPSC targeting approach is more broadly applicable, provided that a relevant functional readout can be identified.

---

## [Decision Letter]

**Decision letter after peer review:**

Thank you for submitting your article "A common *cis-*regulatory variant impacts normal-range and disease-associated human facial shape through regulation of *PKDCC* during chondrogenesis" for consideration by *eLife*. Your article has been reviewed by 3 peer reviewers, and the evaluation has been overseen by a Reviewing Editor and Kathryn Cheah as the Senior Editor. The reviewers have opted to remain anonymous.

Essential revisions:

1) Reviewer 1 made reference to other SNPs potentially in linkage disequilibrium (LD) with rs6740960. Confounding by LD is a well-known problem for the field, which was not addressed in the introductory part of the manuscript, which early on slipped into "causal" language for this SNP at multiple points; this should not be the starting point of this work. Whilst evidence previously presented by Ludwig (HMG 2017) on cleft lip/palate seems informative in this regard, the significance of this work was not highlighted, nor were equivalent imputation data for the facial GWAS signal presented. If further analysis supports that the rs6740960 SNP is "untaggable", the potential mechanisms (ie recurrent mutation vs recombination vs selection) and significance of this should be discussed in the manuscript.

2) Neither the evidence that rs6740960 is the causal SNP, nor if so its target of action (proposed to be PKDCC, rather than the lncRNA gene LINC02898 (c2orf91), which encompasses the rs6740960 SNP), were considered definitive. For example, the resolution of the H3K27HiChip appeared insufficient to establish direct enhancer/promoter interactions, and the H3K27/ddPCR experiment shown in Figure 3D, whilst consistent with differences in the enhancer chromatin state between the SNP alleles, does not indicate the functional effect of that difference. Whilst the homologous region of the mouse genome was not considered to show craniofacial enhancer activity (therefore limiting its use in further experiments), it is unclear why the 500 kb region chosen for the transgenic experiments did not coincide properly with the region that is conserved between vertebrates, thereby limiting the strength of this conclusion (Figure 2B). Such transgenic experiments are subject to position and dosage effects and more numerical details of the number of experiments conducted in each species should be provided.

Overall, the referees agreed that they would like to see a more definitive assay, either in cells or in vivo, that directly demonstrates a functional link between the proposed SNP (and not any other SNP in LD with it) and a downstream readout; and that this readout is specific to PKDCC rather than LINC02898 (c2orf91) (or affects both).

*Reviewer #1 (Recommendations for the authors):*

This is a solid manuscript that is easy to read.

The major unaddressed question here is that all the evidence they present is consistent with the possibility that a SNP in linkage disequilibrium is the actual causal one. This putative SNP could potentially be some distance away and indirectly affect activity of the enhancer that harbors rs6740960. This possibility is not farfetched and needs to be addressed. The strongest test is to engineer to homozygosity in iPSC and then differentiate the cells to chondrocyte. This group has the ability to carry out this incisive experiment. Much easier, and necessary at a minimum, is to carry out standard reporter assays with enhancer constructs bearing the two alleles in their cellular model.

The genome editing experiments were not conducted with sufficient rigor. For instance, it is not mentioned whether the WT unedited cells have undergone clonal isolation in Figure 3C dataset. To negate the possibility of off-target effect of CRISPR-Cas9, data from at least 3 engineered clones and 3 unedited clones are needed. For controls, rather than the parental cells, it is expected to use clones of cells that have been transfected with the targeting construct (or a non-specific targeting construct) and Cas9 but not edited (several such clones are shown in Figure 3, supplement 2). Moreover, a minimum of three clones for unedited and edited versions of the enhancers are necessary for credible quantitation.

The mouse reporter experiments are interpreted with quantitative language:

"The chimp ortholog of the rs6740960 cognate enhancer … relative to the human element, showed both stronger and broader activity, which extended to the medial nasal prominence."

However, lacZ transgenic mouse reporter assays are not quantitative – unless there were single copies, or equivalent numbers of copies, of the two versions of the construct and they were integrated in a safe harbor locus. Insufficient details are provided on how this experiment was performed to know whether this bar was met. There is expression visible in the MNP in both versions, so they cannot argue that this domain is gained in the chimp version of the enhancer. Minimally, the numbers of transgenic embryos scored must be presented. How many embryos that were transgenic but had no signal, or the non-consensus pattern, were scored? The solid conclusion from this experiment is that the enhancer is active in the midface, the conclusion that the chimp and human versions are of different strength is not convincingly established.

The ancestral T allele is found in non-human primates. Do such primates exhibit more protruding jaw morphology than humans (chimps are not the appropriate comparator, because of the other changes in this enhancer)?

*Reviewer #2 (Recommendations for the authors):*

Please address why this enhancer region was not identified as Chimp biased in the original Prescot et al. analysis. Please plot all chromatin signals corrected for input according to Roadmap/ENCODE standards as they can be highly misleading as presented.

Please show publicly available data from Mouse ENCODE for evidence of enhancer activity in mouse craniofacial development. This will allow the reader to put into context the lack of enhancer activity from the lacZ assay for interpreting small differences between human and chimp versions of the sequence. Further test larger regions of the mouse sequence to concretely determine lack of enhancer activity.

The PKDCC knockout mouse has already been well characterized over 10 years ago. While the microCT data is much more quantitative and shows changes in the heterozygote the main premise of the paper is about the regulatory region controlling PKDCC. Since minimal differences in expression were seen in the human system either both copies of the enhancer should be deleted in the human system or perturbation of the endogenous mouse sequence should be attempted.

Please address the potential for a lncRNA directly adjacent to this locus. There is public data to suggest that it is expressed in CNCCs and primary craniofacial tissue but unknown in chondrocytes. Is this RNA species indeed expressed in the cells at hand and do it levels change in a similar way as chromatin activity. Lastly if it is indeed expressed is this RNA itself needed for the expression modulation effect?

Data is in an appropriate repository and available for inspection.

*Reviewer #3 (Recommendations for the authors):*

1: In Figure 2 a multi-species alignment is shown. Since the mouse enhancer seems to behave differently, the question would be how conserved the enhancer is in mouse? Or what is different?

2: Figure 2C: how many embryos showed this exact staining?

---

## [Author Response]

Essential revisions:1) Reviewer 1 made reference to other SNPs potentially in linkage disequilibrium (LD) with rs6740960. Confounding by LD is a well-known problem for the field, which was not addressed in the introductory part of the manuscript, which early on slipped into "causal" language for this SNP at multiple points; this should not be the starting point of this work. Whilst evidence previously presented by Ludwig (HMG 2017) on cleft lip/palate seems informative in this regard, the significance of this work was not highlighted, nor were equivalent imputation data for the facial GWAS signal presented. If further analysis supports that the rs6740960 SNP is "untaggable", the potential mechanisms (ie recurrent mutation vs recombination vs selection) and significance of this should be discussed in the manuscript.

We thank the reviewers for raising this point. We acknowledge that in the original manuscript we have not fully explained the rationale for prioritizing the *rs6740960* SNP for the follow-up experiments. The *rs6740960* SNP emerged as the lead SNP at the 2p21 locus both in our facial GWAS analysis and in a GWAS of susceptibility to clefting by Ludwig (HMG 2017) conducted on an independent European ancestry cohort. The reviewer is correct that other GWAS-significant SNPs in high LD with the lead SNP may be causative. In our GWAS analysis, *rs6740960* is in high LD with only one other genome-wide significant SNP, *rs4952552* (r^2^=0.81 using the 1000 Genomes Project European cohort), located ~16 kb away. To document this, we now present a LocusZoom plot in which LD correlation coefficient r^2^ was computed between *rs6740960* and all other jaw-associated SNPs at the 2p21 locus found from our previous work, along with association significance, the locations of genes, and recombination rates and boundaries (Figure 1—figure supplement 2A). From this plot, it is evident that *rs4952552* is the only genome-wide significant SNP at the locus with the r^2^ value >0.7. Among the two SNPs, we further prioritized *rs6740960*, as we found that this SNP, but not *rs4952552*, overlapped an open chromatin region in facial progenitor cells, CNCCs. We have now included a genome browser snapshot illustrating this to provide the rationale for further focusing on *rs6740960* (Figure 1—figure supplement 2B)*.* Nonetheless, we cannot exclude the possibility that *rs4952552* is associated with regulatory activity in some other, unexamined cell type and therefore might also functionally contribute to the association with jaw shape. In the revised manuscript we have added a sentence: *‘Nonetheless, we note that other SNPs in linkage with rs6740960 may contribute to the observed genetic association of the 2p21 locus with the lower jaw shape.’* Most importantly, through new genome-editing experiments, we now provide direct evidence that *rs6740960* is sufficient to drive chondrocyte-specific changes in *PKDCC* dosage in a native genomic context (please see our response to the second essential point).

Regarding the imputation data: the SNP *rs6740960* was imputed in our GWAS with a high imputation INFO score of 0.91. Imputation was based upon the reference haplotypes from the pre-phased genotypes of European individuals within the 1000 Genomes Project reference panel phase 3. This score values range from 0 to 1 with 0 meaning complete lack of support for a SNP’s allelic phase, and 1 representing SNPs that were genotyped. Higher values indicate higher probabilities of a SNP’s phase based on haplotype configurations within the reference panel. Broadly accepted threshold for inclusion of imputed variants is INFO score > 0.8 at minor allele frequency > 1%. We have now included this information in the methods of the revised manuscript. Of note, *rs4952552,* the SNP in high LD with *rs6740960* was directly genotyped in our GWAS study, which contributes to the high imputation score of the latter variant. Overall, our observations support the ‘taggability’ of the *rs6740960* SNP.

2) Neither the evidence that rs6740960 is the causal SNP, nor if so its target of action (proposed to be PKDCC, rather than the lncRNA gene LINC02898 (c2orf91), which encompasses the rs6740960 SNP), were considered definitive. For example, the resolution of the H3K27HiChip appeared insufficient to establish direct enhancer/promoter interactions, and the H3K27/ddPCR experiment shown in Figure 3D, whilst consistent with differences in the enhancer chromatin state between the SNP alleles, does not indicate the functional effect of that difference. Whilst the homologous region of the mouse genome was not considered to show craniofacial enhancer activity (therefore limiting its use in further experiments), it is unclear why the 500 kb region chosen for the transgenic experiments did not coincide properly with the region that is conserved between vertebrates, thereby limiting the strength of this conclusion (Figure 2B). Such transgenic experiments are subject to position and dosage effects and more numerical details of the number of experiments conducted in each species should be provided.Overall, the referees agreed that they would like to see a more definitive assay, either in cells or in vivo, that directly demonstrates a functional link between the proposed SNP (and not any other SNP in LD with it) and a downstream readout; and that this readout is specific to PKDCC rather than LINC02898 (c2orf91) (or affects both).

In this essential critique point, the reviewers raise multiple concerns regarding the ambiguity of the *rs6740960*’s target gene, the functional effect of *rs6740960* on target gene expression, and the mouse transgenic reporter assays. We address these concerns in turn.

*LINC02898 (C2orf91)* as the potential target gene. *C2orf91 (LINC02898)*, a previously uncharacterized protein-coding gene that was later reclassified as a lincRNA, is indeed the closest gene to *rs6740960* and we have considered it as a potential target of *rs6740960* enhancer activity*.* However, this gene is not expressed at appreciable levels in any of the relevant cell types we have examined, including in vitro derived CNCCs and cranial chondrocytes or fetal facial tissues. For example, in our in vitro derived CNCCs and cranial chondrocytes, we find that *C2orf91* has Transcripts Per Million (TPM) < 1, which is well below the community accepted threshold for defining an “expressed” gene (Figure 3—figure supplement 1). This observation was corroborated in RNA-seq data from CNCCs of (Yankee, J.N. et al. Nat Comm 2023), which were in vitro derived using a different differentiation protocol, as recommended by the reviewers (Figure 3—figure supplement 1). In contrast, in the same cells, *PKDCC* is expressed at levels comparable to those of key CNCC transcription factors, *AP2α* and *NR2F1*. To rule out our RNA-seq processing pipeline as a source of *C2orf91* potentially overlooked higher expression, we note that all RNA-seq datasets in our study were uniformly processed through the ENCODE RNA-seq pipeline to arrive at our observation of TPM < 1 (https://www.encodeproject.org/data-standards/rna-seq/long-rnas/). As the reviewers suggested, we also inspected the *recount3* database of our previously published CNCC dataset (Prescott, S. et al. Cell 2015), but found no more than a few raw reads that aligned to *C2orf91*, which pales in comparison to the hundreds of raw reads that align to *PKDCC* (Author response image 1).

**Author response image 1. sa2fig1:** RNA-seq datasets from in vitro derived late CNCC differentiated from H9 hESCs (from [39]). Datasets were re-processed using *recount3* pipeline recommended by a reviewer. Shown here are the limited number of raw reads that align to lincRNA *C2orf91* (*LINC02898*), versus the hundreds of raw reads that align to *PKDCC*.

We then proceeded to use PCR-based assays to detect *C2orf91* expression. RT-qPCR assays conducted in CNCCs and cranial chondrocytes were unable to distinguish *C2orf91*’s expression from a negative, water-only control sample. Finally, we were unable to detect *C2orf91* expression using ddPCR, which made the study of *C2orf91*’s allelic imbalance upon enhancer perturbation infeasible. Overall, we conclude that *C2orf91* is not expressed at appreciable levels in CNCCs and cranial chondrocytes and thus it is an unlikely – and also technically inaccessible to measure – functional target of the *rs6740960* cognate enhancer in these cell types. In contrast, our data show strong effects of the enhancer deletion (and now also of the *rs6740960* SNP itself) on expression of *PKDCC*, a gene well-expressed in the aforementioned cell types and involved in the relevant biological processes such as chondrogenesis and development of the jaw and palate.

Functionality of the *rs6740960* SNP. In the original manuscript, we demonstrated that the deletion of the *rs6740960* cognate enhancer results in a 40-60% reduction of *PKDCC* expression selectively in cranial chondrocytes. In the revised manuscript, we have now included an additional clonal line with the enhancer deletion to further corroborate these data (one of the reviewers requested three edited lines instead of two) (Updated Figure 3C). All comparisons were done with the isogenic cells that underwent the editing process but remained unedited, and we also note that our measurements are internally controlled, meaning that in the ddPCR we compare *PKDCC* expression from one vs the other allele in the same cell population, making it a highly quantitative and sensitive assay for detecting allelic imbalances.

Nonetheless, as the reviewers pointed out these data do not show that the *rs6740960* SNP is directly driving changes in *PKDCC* expression. To address this critique, we used genome editing to create multiple, independent clonal *rs6740960* A/T heterozygous hESC lines (wild-type H9 hESC used in our studies are A/A homozygous for *rs6740960*). Given that H9 hESCs are heterozygous for other variants at the locus, we obtained cell lines in which one or the other alternate allele was mutated from A to T at *rs6740960* (Figure 3—figure supplement 4). Following differentiation to CNCCs and cranial chondrocytes, we compared allelic imbalance in *PKDCC* expression in these lines to the homozygous, A/A cells. As might be expected, the effects of the single base mutation were not as large as those seen for the deletion of the entire enhancer. Nonetheless, the allele bearing the ‘T’ variant at *rs6740960* showed a 15-30% reduction in *PKDCC* expression across all examined mutant cell lines, and importantly, this allelic imbalance was observed only in chondrocytes (Updated Figure 3C and Figure 3—figure supplement 5). These results definitively demonstrate that *rs6740960* is sufficient to cause chondrocyte-specific changes in *PKDCC* expression.

Of note, while the ‘T’ mutation consistently resulted in diminished *PKDCC* expression across cell lines, the effect was larger when one allele was targeted compared to the other (~30% vs 15% reduction); similar differences were observed in heterozygous deletion lines where enhancer has been deleted from these alternate alleles (Updated Figure 3C, compare haplotype 1 and haplotype 2). These observations suggest that other *cis-*regulatory variants at the locus (not necessarily in LD with *rs6740960*) can modulate effects of *rs6740960* and its cognate enhancer on *PKDCC* expression.

LacZ transgenic reporter experiments in mice. We agree with the reviewers that LacZ transgenic mouse assays are hampered by positional effects of random transgene integration. We have now included Figure 2—figure supplement 1 showing all LacZ-positive transgenic animals bearing the human and chimpanzee enhancer orthologs. These images indicate that while there is a substantial variability in these assays, on average animals bearing the chimpanzee enhancer ortholog do show stronger and broader staining patterns in the frontonasal prominence as compared with the animals bearing the human ortholog (ancestral version). To properly qualify these results, in the revised manuscript we added a sentence:

“Although one caveat to this analysis is that the transgenic reporter was integrated randomly, and thus a subject of positional effects, we have consistently observed weaker activity of the human enhancer across multiple embryos (see Figure 2—figure supplement 1 for all LacZ positive embryos).”

Regarding why the 500 bp region was chosen for the transgenic assays rather than the full vertebrate conserved region: this was for a historical reason, as we first tested the activity of the human and chimp orthologs using the 500 bp region, and subsequently cloned the corresponding mouse region to maintain consistency. However, we do agree with the reviewers that we cannot exclude the possibility that the longer mouse sequence might show activity. Given that our manuscript is focused on the human variant and that the mouse enhancer was tangential to our studies, we decided to remove the mouse ortholog transgenic reporter data or any claims regarding mouse enhancer activity or lack thereof from the manuscript. This in no way impacts our conclusions from the quantitative analyses of the *Pkdcc* dosage on skull shape.